# Low-Level Laser Treatment Induces the Blood-Brain Barrier Opening and the Brain Drainage System Activation: Delivery of Liposomes into Mouse Glioblastoma

**DOI:** 10.3390/pharmaceutics15020567

**Published:** 2023-02-08

**Authors:** Oxana Semyachkina-Glushkovskaya, Denis Bragin, Olga Bragina, Sergey Socolovski, Alexander Shirokov, Ivan Fedosov, Vasily Ageev, Inna Blokhina, Alexander Dubrovsky, Valeria Telnova, Andrey Terskov, Alexander Khorovodov, Daria Elovenko, Arina Evsukova, Maria Zhoy, Ilana Agranovich, Elena Vodovozova, Anna Alekseeva, Jürgen Kurths, Edik Rafailov

**Affiliations:** 1Institute of Physics, Humboldt University, Newtonstrasse 15, 12489 Berlin, Germany; 2Department of Biology, Saratov State University, Astrakhanskaya 82, 410012 Saratov, Russia; 3Lovelace Biomedical Research Institute, Albuquerque, NM 87108, USA; 4Department of Neurology, University of New Mexico School of Medicine, Albuquerque, NM 87131, USA; 5Optoelectronics and Biomedical Photonics Group, Aston Institute of Photonic Technologies, Aston University, Birmingham B4 7ET, UK; 6Institute of Biochemistry and Physiology of Plants and Microorganisms, Russian Academy of Sciences, Prospekt Entuziastov 13, 410049 Saratov, Russia; 7Shemyakin-Ovchinnikov Institute of Bioorganic Chemistry, Russian Academy of Sciences, Miklukho-Maklaya 16/10, 117997 Moscow, Russia; 8Potsdam Institute for Climate Impact Research, Department of Complexity Science, Telegrafenberg A31, 14473 Potsdam, Germany

**Keywords:** blood-brain barrier, infrared low-intensity laser treatment, glioblastoma, liposomes, brain drainage system

## Abstract

The progress in brain diseases treatment is limited by the blood-brain barrier (BBB), which prevents delivery of the vast majority of drugs from the blood into the brain. In this study, we discover unknown phenomenon of opening of the BBBB (BBBO) by low-level laser treatment (LLLT, 1268 nm) in the mouse cortex. LLLT-BBBO is accompanied by activation of the brain drainage system contributing effective delivery of liposomes into glioblastoma (GBM). The LLLT induces the generation of singlet oxygen without photosensitizers (PSs) in the blood endothelial cells and astrocytes, which can be a trigger mechanism of BBBO. LLLT-BBBO causes activation of the ABC-transport system with a temporal decrease in the expression of tight junction proteins. The BBB recovery is accompanied by activation of neuronal metabolic activity and stabilization of the BBB permeability. LLLT-BBBO can be used as a new opportunity of interstitial PS-free photodynamic therapy (PDT) for modulation of brain tumor immunity and improvement of immuno-therapy for GBM in infants in whom PDT with PSs, radio- and chemotherapy are strongly limited, as well as in adults with a high allergic reaction to PSs.

## 1. Introduction

Glioblastomas (GBMs) are the most common central nervous system (CNS) tumour, which can occur at any age [1,2,3]. Small patients with GBMs have a comparably dismal clinical outcome to older patients with morphologically similar lesions [2,3]. Only a few children with GBMs achieving long-term survival despite a variety of therapies applied [2]. However, pediatric and adult GBMs demonstrate biological differences [2].

The removing as much tumor tissue as is safely possible without causing a neurological deficit is the first step in treating adult and small patients with GBMs [2,4,5,6,7]. However, surgery does not much improve the positive outcome. In most cases GBMs recurrence occurs within 2 cm of the resection site [8,9,10]. There is no standard treatment for relapses. Typically, patients who has surgery will receive chemotherapy, which is often combined with the radiation [3,4,8,11]. Despite differences between the adult and pediatric populations, therapy for pediatric GBMs are often assigned based on data from adult GMBs without consideration for the biological differences in the GBM type [3]. However, there are several limitations in the GBM therapy in children. So, radiotherapy has not consistently been proven to be statistically beneficial for children with GBMs [11,12]. In newborns, radiotherapy can result in a severe impairment of the brain development, growth and cognitive abilities [13] due to direct ionization of DNA causing not only DNA degradation in the GBM cell nuclei but also DNA mutations in healthy neuronal cells surrounding GBM [14]. Chemotherapy in kids is still considered as controversial and is not universally accepted because this method would have intrinsic toxicity to the whole organism [15,16].

The blood-brain barrier (BBB) limits the delivery of vast majority of cancer therapeutics that creates challenge of pharmacological therapy of GBMs, especially treatment of satellite tumor areas that grow along healthy cerebral vessels with an intact BBB [17,18,19,20,21]. Therefore, the effective therapy of recurrent GBMs depends on the development strategies of bypassing an intact BBB at the GBM border to prevent tumor migration and progression [18].

The photodynamic therapy (PDT), including intracavitary PDT and interstitial PDT (iPDT), is considered as a promising least-toxic tool in treating both pediatric and adult GBMs [22,23,24,25,26,27,28,29]. The intracavitary PDT is performed after surgical resection of primary GBM, while iPDT is used for mini-invasive therapy of recurrent GBM, where semiconductor lasers are positioned inside the surgical cavity or directly into new tumor mass [28,29]. PDT combines a light source and photosensitizers (PSs). United States Food and Drug Administration approved 5-aminolevulinic acid (5-ALA) for PDT of GBMs [30]. The mechanisms of the anti-cancer PDT effect is single oxygen-induced microvasculature collapse leading to death of tumor cells [26,31]. Recent studies have been discovered the new vascular effects PDT-5ALA associated with opening of the BBB (BBBO) and activation of the brain drainage system [32,33,34,35,36,37,38,39,40]. PDT-BBBO is considered as the new niche in the development of innovative pharmacological strategies of modulation of brain tumor immunity and improvement of immuno-therapy for GBM [41]. However, PDT has limitations to use in small children and infants as well as in adult patients with allergy to PSs [42,43,44,45]. Therefore, the development of new PSs-free PDT can give new opportunities to overcome the limitations of PDT-therapy of GBM in such patients.

Recently developed compact and efficient quantum-dot lasers emitting in the near-infrared spectral range centered at around 1268 nm offer such a promising opportunity. This wavelength irradiation is capable to directly generate oxygen photoexcitation in complex media, leading to a generation initially singlet oxygen (^1^O_2_) and other reactive oxygen species (ROS) without the need for PSs and with good penetration into the brain (up to 6 mm via the intact skull) [46,47,48,49].

It was believed that the ^3^O_2_–^1^O_2_ transition in molecular oxygen is prohibited based on spinorbital selection rules. However, new experimental data warrant revision of this point of view. The enhancement of the ^3^O^2^–^1^O_2_ transition has been attributed to the major intensity contribution from O^2^–O^2^ bimolecular collisions, which mix electron orbital states by intermolecular exchange interaction, introducing some allowed characters into previously forbidden transitions. However, direct monitoring of ROS in the tissues remains technically challenging. Nevertheless, direct ROS (e.g., ^1^O_2_ or O_-2_) formation by oxygen photoexcitation with 1265 nm or 1064 nm has been shown in several experiments in vitro, including ours on cancer cells [46,50].

Based on novel fundamental data about ^1^O_2_-mediated BBBO by PDT and the emergence of new perspective approaches for direct ^1^O_2_ generation using low level laser treatment (LLLT) with the laser1268 nm, we studied LLLT-induced BBBO and activation of the brain drainage system for the delivery of liposomes into the brain and the GBM tissues. We also analyzed the mechanisms underlying LLLT-BBBO, including the expression of tight junction (TJ) proteins, neuronal metabolic activity, measurement of the transendothelial resistance (TEER), and the ^1^O_2_ generation in the in vitro BBB model.

## 2. Materials and Methods

### 2.1. Subjects

Male C57BL/6 mice (25–28 g) were used in all experiments and were obtained from the National Laboratory Animal Resource Centre in (Pushchino, Moscow, Russia) or Jackson Laboratory (Bar Harbor, ME, USA). The animals were housed under the standard laboratory conditions, with access to food and water ad libitum. All experimental procedures were performed in accordance with the “Guide for the Care and Use of Laboratory Animals”, Directive 2010/63/EU on the Protection of Animals Used for Scientific Purposes, and the guidelines from the Ministry of Science and High Education of the Russian Federation (№ 742 from 13.11.1984), which have been approved by the Bioethics Commission of the Saratov State University (Protocol No. 7) and the Institutional Animal Care and Use Committee of the University of New Mexico, USA (19-200767-HSC200247).

### 2.2. Laser 1122 nm and 1268 nm Irradiation

The fiber Bragg grating wavelength locked high-power laser diodes (LD-1268 or 1122-FBG-350, Innolume, Dortmund, Germany, laser driver: THORLABS CLD1015) emitting at 1268 nm or 1122 nm were used as a source of irradiation. The laser (1268/1122 nm) output power was adjusted from 10 to 90 mW by the setting of the laser diode current. The laser driver CLD1015 (THORLABS, Newton, NJ, USA) was controlled with a personal computer by homemade software developed with LabVIEW (National Instruments, Austin, TX, USA). The software was used to program the dose and stimulation timing. The procedure of turning the laser on and off and setting the laser diode output power was performed automatically. Maximal light intensity at the skull surface complies with laser safety standards.

The laser irradiation was pigtailed with a single-mode optical fiber with an FC connector. The additional single-mode patch cord was attached to guide laser irradiation toward the animal head. The outer sleeve of the patch cord was stripped out, and a distal end of the optical fiber was mounted into a centering inset of a head mount and polished (Figure 1). The head mount was 3D printed with the polylactic acid (Villacryl S, Zhermack, Italy). It consisted of a base with an internal M6x1 thread attached to an animal head with the dental acrylic cement (Stoelting Co., Wood Dale, IL, USA) and an optical fiber holder. Thus, the fiber holder can be easily connected to the base prior to laser stimulation, and no anesthesia is to be required for animal. The fiber holder allows free rotation of the centering inset to avoid optical fiber twisting when the animal moves. The holder length was selected to ensure a 5 mm illuminated area at the skull surface as a result of light divergence over the fiber acceptance angle. The overall length of the holder was 35 mm to protect the optical fiber from the animal.

For in vitro studies, the laser irradiation power was 70 mW to ensure the same laser radiation intensity over the BBB cells as for in vivo experiments. The 70 mW laser radiation power applied to the mouse skull corresponds to the intensity of 70 × 0.35/(1.2 × 0.2) = 102 mW/cm^2^ at the brain surface. Thus to ensure the same intensity at the BBB cells in lunule irradiated with 5 mm diameter collimated beam and assuming 5% attenuation because of reflection and absorption of laser radiation passing through the bottom of lunule the reduced laser radiation power was set to 102 mW × 0.2 cm^2^/0.95 = 21 mW, where 0.95 is the transmission coefficient of plastic lunule. Afterward, the BBB cells were irradiated following the protocol comprising 17 min laser 1268 nm + 5 min pause + 17 min laser 1268 nm + 5 min pause + 17 min laser 1268 nm the total irradiation time was 51 min and the dose was 312 J/cm^2^ that corresponds to the effect of 70 mW laser radiation power applied in vivo through the skull.

During both in vivo and in vitro experiments laser irradiation power was measured with power meter (1815-C with thermopile sensor head 818P-070-20, Newport Inc., Irvine, CA, USA).

### 2.3. Spectrofluorometric Assay of the Evans Blue Extravasation

Before or 1 h/4 h/24 h after the laser 1268 nm irradiation, the Evans Blue dye (Sigma Chemical Co., St. Louis, Missouri, 2 mg/25 g mouse, 1% solution in physiological 0.9% saline) was injected into the femoral vein and circulated in the blood for 30 min. Then, the mice were decapitated, their brains were quickly collected and placed on ice (no anticoagulation was used during blood collection). Prior to the brain removal, it was perfused with saline to wash out the remaining dye in the cerebral vessels. The Colorimetric measurement level of the Evans Blue albumin complex (EBAC) in the brains was carried out at a wavelength of 640 nm using a Cary Eclipse fluorescent spectrophotometer (Agilent, Santa-Clara, CA, USA) in accordance with the recommended protocol [51].

### 2.4. In Vivo Real-Time Two-Photon Laser Scanning Microscopy (2PLSM)

The BBB permeability was continuously monitored via an optical window [52] by measuring the perivascular tissue fluorescence of fluorescein isothiocyanate–dextran (FITCD) 70 kDa (Sigma-Aldrich, St Luis, MO, USA, in saline 5% wt/vol) in 10 mice at different time points: before or 1, 4 and 24 h after the 1268 nm or 1122 nm lasers exposure as described previously with some modifications [53]. During the imaging, mice were kept under inhalation anesthesia with 1% isoflurane at 1 L/min N_2_O/O_2_–70:30. The body temperature was maintained at 37.5 °C by a homoeothermic blanket system with a rectal probe (Harvard Apparatus, Holliston, MA, USA). The brain temperature was kept at 37 °C using an objective heating system with a temperature probe (Bioptechs Inc., Butler, PA, USA). The FITCD was injected through the tail vein (∼100 μL) at an estimated initial concentration in the blood serum of 150 μM. Imaging was done using the Prairie View Ultima system with Olympus BX51WI upright microscope and water-immersion LUMPlan FL/IR 20×/0.50 W objective. The excitation was provided by a Prairie View Ultima multiphoton laser scan unit powered by a Millennia Prime 10 W diode laser source pumping a Tsunami Ti: sapphire laser (Spectra-Physics) tuned up to 810 nm central wavelength. Fluorescence was band-pass filtered at 510–550 nm. Real-time images (512 × 512 pixels, 0.15 μm/pixel in the x- and y-axes) were acquired using the time series mode of the Prairie View software (30 s intervals between images, 30 min sessions). Between the sessions mice were awake and were kept at the animal facility. In offline analyses using the NIH ImageJ, the BBB permeability was evaluated by measuring changes in perivascular tissue fluorescence in planar images of the cortex taken 50 and 150 µm depth in 20 min after the FITCD injection, as described previously [53]. For each image, an intensity of fluorescence of ten randomly chosen regions of interest over the vessel areas and 10 corresponding perivascular brain parenchyma areas were evaluated. The obtained estimates in the interstitial space were normalized to the maximal fluorescence intensity in the blood vessels and expressed as a percentage of fluorescence intensity (modified technique from [53]). The estimated quantity of fluorescein that leaked out of the vessel and remained in the vessel for 20 min after the injection was calculated from the initial vascular fluorescence equal to ~150 µM of fluorescein. The time courses of fluorescein leakage for each group were also plotted.

### 2.5. Magnetic Resonance Imaging of the BBB Permeability

The MRI was conducted on the same mice, which we used for 2PLSM, at different time intervals 0–before and 1, 4, 24 h after the 1268 nm or 1122 nm lasers exposure on a 7-T dedicated research the MRI scanner (Bruker Biospin; Billerica, MA, USA). Signal transmission and detection were done with a small-bore linear RF coil (inner diameter of 72 mm) and a single-tuned surface coil (RAPID Biomedical, Rimpar, Germany). The mice were kept under inhalation anesthesia (2% isoflurane at 1 L/min N_2_O/O_2_–70:30). Respiration and heart rate were monitored during the MRI measurements, and the body temperature was maintained at 37.0  ±  0.5 °C. Anatomical T2-weighted images were acquired before each the BBB permeability determination with a fast spin-echo sequence (rapid acquisition with relaxation enhancement (RARE)) (Repetition Time (TR)/Echo Time (TE)  =  5000 ms/56 ms, Field of View (FOV)  =  4 cm  ×  4 cm, slice thickness  =  1 mm, slice gap (inter-slice distance)  =  1.1 mm, number of slices  =  12, matrix  =  256  ×  256, number of averaging  =  3) as previously described [54].

To non-invasively evaluate the BBB permeability, we used a modified dynamic contrast-enhanced (DCE)-MRI and graphical analysis of the resultant image data [54]. Mice were injected 0.1 mM/kg of gadolinium-diethylene-triamine-pentaacetic acid (Gd-DTPA, MW= 938 Da; Bayer Healthcare) as a bolus injection in the tail vein, followed by imaging. DCE-MRI was performed using a transverse fast T1 mapping that consisted of obtaining pre-contrast (three sequences) and post-contrast (15 sequences) images up to 15 min after the contrast injection. The details of the pulse sequence T1 RARE for T1 weighted imaging are: FOV  =  2 cm  ×  2 cm, slice thickness  =  1.5 mm, slice gap  =  0, matrix size  =  128  ×  128, TR/TE = 377 ms/12.3 ms, number of averages  =  2, total scan time  =  48 s 25 ms.

### 2.6. MRI Data Analysis and Processing

The method is based on the leakage of the contrast agent from the blood plasma into the brain tissues through the BBB, resulting in a change in the MR signal intensity. The rate of changes in the MRI signal intensity relates to the BBB permeability (Ki). The T1 map was reconstructed with the t1epia fitting function in the Bruker ParaVision Image Sequence Analysis (ISA) tool. Previous research has demonstrated that the blood-to-tissue transfer or influx constant, Ki could be obtained by a graphical analysis of timed series of tissue and arterial concentrations of a contrast agent [54]. Since the contrast agent concentration is proportional to changes of 1/T1(Δ(1/T1(t))), the color-coded map of Ki was constructed from repeated estimates of Δ(1/T1(t)), where pixels with higher intensity color represent higher the BBB permeability. A custom-made computer program in MATLAB (Mathworks, Natick, MA, USA), which implemented the above principle, was used to generate the Ki map.

### 2.7. The In Vitro BBB Model

The experiments were performed on the in vitro model of BBB consisting of a mixed culture of endothelial, astroglial and neural cells. Isolation and preparation of the primary brain microvessels (BMV) culture were performed according to the protocol of Liu et al. [55]. The BMVs were phenotyped with monoclonal antibodies to the endothelial marker the zonula occludens-1 (ZO1) using a standard immunohistochemistry protocol using primary anti-ZO1 antibodies (Santa Cruz Biotechnology, Santa Cruz, CA, USA) and secondary antibodies labeled with Alexa Fluor 488 (Abcam, Cambridge, United Kingdom) followed by an Olympus FV10i-W confocal laser scanning microscope (Olympus, Tokyo, Japan). Astroglial and neural cells for the in vitro BBB model are derived from embryonic neurospheres as described in Ref. [56]. The TEER was measured using an EVOM2 volt-ohmmeter with the STX-2 electrodes (World Precision Instruments, Sarasota, FL, USA). Resistance values (Ωcm^2^) were corrected by subtracting the resistance of an empty Transwell filter.

### 2.8. Measurement of Lactate Levels in the Culture Medium

The BBB model was formed in a 24-hole tablet. A day later, they were exposed to the 1268 nm laser. An hour after exposure, the medium was taken and frozen at −80 °C to determine the lactate level. The measurement was carried out by the spectrocolorimetric method according to the standard Abcam protocol using the L-Lactate Colorimetric/Fluorometric Assay Kit (ab65330, Abcam, Cambridge, UK). The reaction mixture contained 46 µL Lactate Assay Buffer, 2 µL Lactate Probe, 2 µL Lactate Enzyme Mix and 100 µL of the test sample. The reaction mixture was incubated for 30 min at room temperature, after which a colorimetric measurement was performed at a wavelength of 570 nm using a Cary Eclipse fluorescent spectrophotometer (Agilent, Santa-Clara, USA). The lactate concentration in the studied samples was measured by a calibration curve, which was plotted at 6 points with a lactate concentration of 0–10 pmol/µL. To construct a calibration curve, the reaction mixture was added 0, 6, 12, 18, 24, 30 mL of a standard lactate solution at a concentration of 100 nmol/mL was incubated for 30 min at room temperature, after which a colorimetric measurement was performed at a wavelength of 570 nm using a CM 2203 spectrofluorimeter (Solar, Minsk, Belarus).

### 2.9. Implantation of EPNT-5-TagRFP GBM

Mice were deeply anesthetized with intraperitoneal Zoletil (Virbac, Carros, France) in a dose of 20 µg/kg, moved into a stereotaxic head holder, and immobilized on the stereotactic system (51500 Stoelting, Wood Dale, IL, USA) by fixation of the head. The scalp of the anesthetized mice was shaved and scrubbed with betadine (EGIS, Budapest, Hungary) 3 times, followed by an alcohol rinse. An incision was made over the sagittal crest from the bregma to the lambdoid suture, and the periosteal membrane was removed. A small dental drill (Foredom, Moscow, Russia) was used to create a burr hole (1.3 mm in diameter) through the bone without tearing the dura mater in the exposed cranium 1 mm anterior and 1 mm lateral to the bregma.

The cell line EPNT-5 of mouse GBM was obtained from the Russian Cell Culture Collection of Vertebrates, Institute of Cytology, St. Petersburg, Russia). EPNT-5- cells were cultured in Dulbecco’s modified eagle medium (DMEN, PanEco, Moscow, Russia) containing 10 % fetal calf serum (Biosera, Rue Lacaille, France), 4 mM glutamine (PanEco, Moscow, Russia), penicillin (50 IU/mL, PanEco, Moscow, Russia) and streptomycin (50 mg/mL) (PanEco, Moscow, Russia). Versen’s solution (PanEco, Moscow, Russia) with the addition of 0.25% trypsin (Gibco, New York, NY, USA) was used to remove cells from the surface of the culture plastic. The cells were cultured in a humid environment in a CO_2_ incubator at a temperature of 37 °C, with 5% CO_2_. The number of the GBM cells was counted on the TC20 Bio-Rad automatic cell counter (Bio-Rad, Moscow, Russia), and the viability analysis was determined using a trypan blue dye. The mouse EPNT-5 of GBM cells were transfected with pTagRFP-C DNA plasmid using the published method of liposomal transfection [57,58] followed by selection using geneticin (G418 antibiotic, neomycin analog). The resulting cell line EPNT-5-TagRFP has stable cultural and morphological characteristics.

The GBM cells (5 × 10^5^ cells per mouse) were injected at a depth of 1 mm from the brain surface with a Hamilton microsyringe (Hamilton Bonaduz AG, Switzerland) in a volume of 7 µL at a rate of 0.5 µL/min. Afterward, the burr hole was sealed with sterile bone wax (Unisur, Karnataka, India) and tissue glue (Ethicon, St. Louis South, MO, USA), the wound was sutured closed with 3-0 absorbable suture material, and treated with 2% brilliant green solution (ElaDum Pharma Ltd., Kishinev, Moldova). The mice were removed from the stereotaxic head holder (Stoelting, Wood Dale, IL, USA), given with water 0.01 mg/kg ibuprofen (Synthesis, Kurgan, Russia) and 50 K bicillin (Synthesis, Kurgan, Russia), i.m., returned to their cages after recovery in the temperature-controlled recovery cage and moved back to the animal facility after recovery.

### 2.10. Synthesis of GM_1_-Liposomes

Fluorescently labeled GM_1_-liposomes were prepared as described earlier [37]. Briefly, a lipid film obtained from egg yolk phosphatidylcholine (Lipoid GmbH, Ludwigshafen, Germany)—ganglioside GM_1_ from bovine brain (Sigma Chemical Co., St. Louis, MO, USA), 9:1 (by mol.), and 1 mol. % the fluorescent lipid probe BODIPY-phosphatidylcholine synthesized as described earlier [33] (λ_ex_ = 497 nm, λ_em_ = 505 nm) was hydrated in physiological saline (phosphate buffer with 0.5 mM EDTA, pH 6.8; total lipid concentration 25 mM). The resulting suspension was subjected to seven cycles of freezing/thawing (liquid nitrogen/+40 °C) and extruded 10 times through polycarbonate membrane filters (Nucleopore, Sigma-Aldrich, St. Louis, MO, USA) with a 100 nm pore diameter using an Avanti Mini-extruder (Northern Lipids, Burnaby, BC, Canada). Particle size was measured by dynamic light scattering with the Brookhaven equipment (Brookhaven Instruments Corp. 90 Plus Particle Sizing Software ver. 4.02, Holtsville, NY, USA) in at least three runs per sample: effective diameter and polydispersity indexes were 99.6 nm and 0.16, respectively. The Zeta potential of liposomes of this composition measured using a ZetaPALS analyzer (Brookhaven Instruments Corp., Holtsville, NY, USA) was −48.3 ± 0.9 mV [59].

### 2.11. Analysis of the ROS Production

The level of intracellular ROS was assessed fluorometrically by a dihydroethidium (DHE) approach [60]. This assay is used to detect active ^1^O_2_ radicals in cells in the presence of other ROS. A 30 mM DHE solution was prepared by dissolving 10 mg of DHE (95%, Sigma Aldrich, St. Louis, MO, USA) in 1 mL of Dimethyl sulfoxide (Sigma Aldrich, St. Louis, MO, USA). Cells were seeded in glass-bottom Petri dishes to form the BBB. After 24 h, DHE was added to the cocultures in an amount of 1 μL per 1 mL of medium and incubated for 30 min. Control plates were washed with nutrient medium after 30 min and imaged with a confocal microscope (λ_ex_ = 540 ± 25 nm; λ_em_ = 600 ± 40 nm) for 10 min (Time Laps). The experimental plates were washed with nutrient medium, irradiated with the 1268 nm laser and immediately examined using a confocal microscope under the same conditions. Confocal microscopy was performed using a fully automated confocal laser scanning microscope with water immersion Olympus FV10i-W (Olympus, Tokyo, Japan). Fluorescence level and quantitative data of fluorescence intensity were calculated by FV10-ASW 4.0 microscopy software (Olympus, Tokyo, Japan).

### 2.12. Optical Monitoring of the Brain Drainage System and Liposomes Delivery into GBM

For the study of photo-activation of the brain drainage system, we investigated distribution of FITCD on dorsal and ventral parts of the brain after intraventricular injection of tracer in mice treated and not by the 1268 nm laser irradiation. Ten days before experiments, a polyethylene catheter (PE-10, 0.28 mm ID × 0.61 mm OD, Scientific Commodities Inc., Lake Havasu City, Arizona, United States) was implanted into the right lateral ventricle (AP—1.0 mm; ML—1.4 mm; DV—3.5 mm) for injection of FITCD according to the protocol reported by Devos et al. [61]. An amount of 5 μL of FITCD 70 kDa (Sigma-Aldrich, St. Luis, MO, USA) at a rate of 0.1 μL/min was injected into the right lateral ventricle in two groups of awake healthy mice, including animals treated with the 1268 nm laser irradiation and intact mice without the 1268 nm laser irradiation. The cerebral vessels were filled with the Evans Blue dye (2 mg/100 g, 1% solution in physiological 0.9% saline, Sigma-Aldrich, St. Louis, MO, USA) that was injected intravenously via the tail vein. The FITCD and Evans Blue were injected 30 min before confocal and multiphoton microscopy.

To monitor accumulation of liposomes in the GBM tissue, liposomes in physiological 0.9% saline (0.2 mL/100 g, i.v.) were intravenously injected via the tail vein 30 min before confocal imaging [37].

The ex vivo optical visualization of the distribution of FITCD in the brains or accumulation of liposomes in the GBM tissues was performed with an A1R MP confocal microscopy system based on the Ni-E focusing nosepiece upright microscope (Nikon, Tokyo, Japan). Three laser excitation sources with 488 nm (for FITCD), 640 nm (for the Evans Blue dye) and 561 nm (for liposomes) wavelengths and a set of photomultipliers as detectors were used for multichannel confocal imaging with 2× and 10× objectives. Fresh samples of whole brains were immersed in saline solution in a Petri dish and covered with horizontally oriented cover glass. The Petri dish was placed on an electromechanical stage of the confocal imaging system. The images were obtained using NIS-Elements software (Nikon Instruments Inc., Tokyo, Japan) and analyzed using Fiji software (Open-source image processing software 2.9.0) and Vaa3D (Open Source visualization and analysis software v0.6007).

### 2.13. Immunohistochemical (IHC) Assay

For the IHC analysis, the brain tissue without and 1 h, 4 h and 24 h after the laser 1268 nm irradiation was collected and free-floating sections were prepared. Mice were decapitated and the brains were quickly removed, fixed with 4% neutral buffered formalin for 24 h, and cut into 50-µm thick slices on a vibratome (Leica VT 1000S Microsystem, Wetzlar, Germany). The antigen expression was evaluated on sections of the mouse brain, according to the standard method of simultaneous combined staining of the drug (abcam protocols for free-floating sections). The nonspecific activity was blocked by 2-h incubation at room temperature with 10% BSA in a solution of 0.2% Triton X-100 in PBS. Solubilization of cell membranes was carried out during 1-h incubation at room temperature in a solution of 1% Triton X-100 in PBS. Incubation with primary antibodies was overnight at 4 °C: Neural/glial antigen 2 (NG2) and Glial fibrillary acidic protein (GFAP) (1:500; Abcam, Cambridge, UK); Junctional adhesion molecule (JAM), P-glycoprotein (Pgp), Cluster of Differentiation 31 (CD31), Zonula occludens-1 (ZO-1), Claudin-5 (CLDN5), Monocarboxylate transporter 1 (MCT1) (1:250, Abcam, Cambridge, UK). At all stages, the samples were washed 3–4 times with 5-min incubation in a washing solution. After that, the corresponding secondary antibodies were applied goat anti-mouse IgG (H+L) Alexa Four 488 and goat anti-rabbit IgG (H+L) Alexa Four 555; (Invitrogen, Molecular Samples, Waltham, MA, USA). The monoclonal antibodies Alexa Fluor 488 and Alexa Fluor 555 were used in a 1:1000 dilution in PBS with 1% BSA. At the final stage, the sections were transferred to the glass and 15 µL of mounting liquid (50% glycerin in PBS) was applied to the section. The preparation was covered with a cover glass and confocal microscopy was performed.

Sections of mice brain slice were visualized using a confocal microscope (Nikon A1R MP, Nikon Instruments Inc., Tai Seng, Singapore) with a ×20 lens (0.75 NA) or a ×100 lens for immersion in oil (0.45 NA). Alexa Fluor 488, Alexa Fluor 555 and EBAC were excited with excitation wavelengths of 488 nm, 560 nm and 640 nm, respectively. Three-dimensional visualization data was collected by obtaining images in the x, y and z planes. The images were obtained using NIS-Elements software (Nikon Instruments Inc.) and analyzed using Fiji software (Open-source image processing software 2.9.0) and Vaa3D (Open Source visualization and analysis software v0.6007).

### 2.14. Histological Analysis of the Brain Tissues

All mice were euthanized with an intraperitoneal injection of a lethal dose of isoflurane. Afterward, the brains were removed and fixed in 10% buffered paraformaldehyde. The paraformaldehyde-fixed specimens were embedded in paraffin, sectioned (4 µm) and stained with haematoxylin and eosin. The histological sections were evaluated by light microscopy using the digital image analysis system MicroVisor medical μVizo-103 (LOMO, Saint Petersburg, Russia).

### 2.15. Statistical Analysis

The results are presented as mean ± standard error of the mean (SEM). Differences from the initial level in the same group were evaluated by the Wilcoxon test. Intergroup differences were evaluated using the Mann–Whitney–Wilcoxon test. The significance levels were set at *p* < 0.05 for all analyses.

## 3. Results

### 3.1. Selection of Optimal Dose of Laser Irradiation for Effective BBBO

To define the optimal and safe parameters of the 1268 nm irradiation for BBBO, we tested the different optical intensities of the laser irradiation on the BBB permeability and evaluated the time window of the BBB opening/closing. For this aim, we performed a quantitative analysis of the BBB permeability to the EBAC (68.5 kDa) before (0 h) and 1-4-24 h after the 1268 nm irradiation of different intensities (from 10 to 90 mW with 10 mW steps). The laser intensity and time of application on a selected area of the brain were set by random selection. Here we presented 2 time points–1 h after the 1268 nm application, when we observed the significant changes in the BBB leakage, and 4 h after the 1268 nm irradiation, when we observed the BBB recovery.

The spectrofluorometric analysis revealed a laser dose-dependent BBBO to the EBAC 1 h after irradiation (Appendix A). So, 1 h after the 1268 nm laser application (ranging 10–70 mW) to the mouse head through the intact scalp and the skull, the BBB permeability to the EBAC was a 2.72–10.63-fold higher compared to the control group without the laser irradiation. Further increase in the intensity of the laser irradiation to 80 mW and 90 mW did not show an increase in the EBAC leakage, which was 6.63-fold and 6.81-fold higher than in the control group. We demonstrated similar results in our previous study using PDT-induced BBBO [38]. The 1268 nm-laser intensity in the range of 10–70 mW caused an increase in the BBB permeability in inconsistent reproducibility (30–60% of mice), while the 1268 nm-laser intensity of 70–90 mW induced BBBO in all tested mice (Appendix A).

Our most important findings here were that 4 h after the laser 1268 nm irradiation, the BBB permeability to the EBAC completely recovered and was intact at least during the next 24 h.

To exclude possible the heat effects generated by the laser, we monitored the temperature in the brain tissue and in the skin of the mouse head without and after the laser 1268 nm application (10–90 mW). The laser intensity in a range from 10 to 50 mV did not change the skin temperature, while 60–90 mW increased the skin temperature by ~2 °C. However, no effect of the laser irradiation with the different tested doses was observed on the brain temperature (Appendix A).

The histological analysis of the brain tissues did not show any morphological changes neither 1 h nor 4 h after the laser 1268 nm irradiation (70–90 mW) (Appendix A). The possible explanation is a scattering effect of the skin and the skull that significantly decreases the laser intensity. Since 1268 nm (70 mW) was accompanied by a maximal increase in the BBB permeability to the EBAC in all mice and these changes were not associated with morphological changes in the brain tissues as well as in the temperature on the brain surface, in the further studies we used this laser intensity as optimal for BBBO.

### 3.2. 1268 nm Laser Irradiation Induces Singlet Oxygen Generation in the In Vitro BBB Model

In our previous work, we clearly demonstrated that a single laser 1268 nm pulse triggers dose-dependent ^1^O_2_ generation in both normal keratinocytes and HeLa cancer cells [50]. Since the detection of ^1^O_2_ in living systems is still not well recognized, in this work we used DHE that is oxidized to dihydroxyethidium (DHOE) by the superoxide anion (O_2_^−^), the first by-product of ^1^O_2_ reduction and ROS precursor in the cell. DHOE fluoresces at a different wavelength (Em = 595 nm) than DHE (Em = 415 nm) and thus may be used to visualize superoxide production.

Considering our previous data, in these experiments, we also used DHE for the detection of ^1^O_2_ using the in vitro BBB model before and after the laser 1268 nm irradiation (70 mW) that was established as optimal for BBBO.

Our results emphasize that the fluorescence of oxidized DHE fluorescence was 2.7-fold higher after the 1268 nm laser irradiation than the control untreated group (1437.73 ± 63.40 a.u. vs. 514.48 ± 56.41 a.u., *p* < 0.001, the Wilcoxon, Mann-Whitney U tests) (Appendix A).

Thus, the 1268 nm laser irradiation induces ^1^O_2_ generation in the endothelial cells and the astrocytes, which might be a trigger mechanism for BBBO. To check this hypothesis, we analyzed the BBB permeability to low- and high-molecular-weight substances in the in vivo experiments using the laser 1268 nm, which generates ^1^O_2_, and the laser 1122 nm laser, which does not produce ^1^O_2_.

### 3.3. Laser Irradiation with 1268 nm but Not with 1122 nm Induces BBBO

We performed an MRI imaging of the BBB permeability to Gd-DTPA (938 Da) with a custom-made sequence (Ki values) for a post hoc evaluation of changes in the MRI signal intensity. Figure 2a shows T2 anatomical scan of the mouse brain representing the coordinates for rapid T1 for Gd-DTPA scanning. Figure 2b demonstrates the pseudocolored Ki map image of the mouse brain with intact BBB (IBBB) before the 1268 nm irradiation. The data presented in Figure 2c illustrates a significant (*p* < 0.01) increase in the BBB permeability to Gd-DTPA in the cortex and sub-cortical area of the same brain (3 mm depth, brighter–more permeable) 1 h after the laser 1268 nm irradiation and nearly IBBB after irradiation with the 1122 nm laser (Figure 2d). The quantitative analysis obtained from Ki maps of the BBB permeability to Gd-DTPA is presented on Figure 2e, which reflects the maximal Gd-DTPA leakage 1 h after only the laser irradiation with 1268 nm but not with 1122 nm. The average Ki map values reached 79.6 ± 15 a.u. (*p* < 0.01) only 1 h after irradiation with laser 1268 nm. All other Ki values for lasers 1268 nm and 1122 nm were comparable to baseline–33.8 ± 9.0 a.u. The recovery of the BBB permeability was observed 4 h after the 1268 nm irradiation which confirmed our ex vivo data with the use of the EBAC suggesting reversible effects of the 1268 nm laser on the BBB permeability.

In vivo imaging of FITCD 70 kDa leakage into the brain parenchyma with 2PLSM confirmed our MRI results. Indeed, 2PLSM showed BBBO 1 h after the laser 1268 nm irradiation but not after the 1122 nm laser exposure (Figure 2f–m). As it shown on Figure 2f, just after the FITCD intravenous injection in mouse irradiated with the laser 1268 nm (1 h after laser exposure), the cerebral vessels filled with FITCD were clearly seen over the dark background of unstained tissue. Twenty minutes later, increased perivascular fluorescence due to FITCD leakage from microvessels into the brain parenchyma, indicating BBBO, was observed in the same area (Figure 2g). In the mouse brain that was irradiated with the laser 1122 nm, the FITCD extravasation was not observed (Figure 2h). Quantification of the vascular permeability at 20 min after the FITCD injection in mice irradiated with the laser 1268 nm presented in Figure 2i that shows an increase to 17.4 ± 4.2% 1 h (*p* < 0.01); and to 7.2 ± 4% 4 h after the laser 1268 nm irradiation (*p* = 0.15) compared with to the baseline (4.4 ± 1.2%). In mice irradiated with the laser 1122 nm, no significant difference in the FITCD leakage was found (4.37 ± 1%, 5.89 ± 2.9% and 5.1 ± 4% for the baseline, 1 h and 4 h after the laser irradiation, respectively) (Figure 2i).

Time series plots of perivascular fluorescence of FITCD illustrate the dynamics of dye extravasation that reached a plateau at ~12 min after the FITCD injection in mice irradiated with the laser 1268 nm (Figure 2j). There was not any changes in the dynamics of the FITCD fluorescence in animals irradiated with the laser 1122 nm (Figure 2k).

Calculated maximum FITCD concentration in the blood and in the brain parenchyma revealed the reduction of FITCD level from 144.2 ± 19.8 µM to 104.0 ± 17.4 µM in the blood plasma and an increase of dye content from 4.8 ± 2.9 µM to 28.4 ± 6.1 µM in the brain perivascular parenchyma in mice after the 1268 nm laser irradiation (Figure 2l). In mice irradiated with the laser 1122 nm, FITCD in the brain parenchyma and in the blood plasma were similar at all-time points (Figure 2m).

Thus, both in vivo MRI and 2PLSM clearly demonstrated that laser 1268 nm (^1^O_2_ and ROS generation) but not laser 1122 nm (no ^1^O_2_/ROS generation) induced reversible BBBO.

### 3.4. Mechanisms of Underlying 1268 nm-Induced BBBO and Recovery

To study the mechanisms responsible for the 1268 nm-induced BBBO and recovery, we analyzed the expression of the BBB machinery proteins as well as an indicator of the BBB integrity–TEER 1 h after the 1268 nm application, when the BBB was opened and in the next day, when the BBB was closed.

Figure 3a,b demonstrates the confocal imaging of IBBB and the 1268 nm laser-induced BBBO to the EBAC and dye distribution among astrocyte end-feets, suggesting the effective penetration of the dye into the brain tissue and crossing the BBB. The laser-mediated BBBO was accompanied by an increase in the expression of Pgp, an ATP-dependent drug transport protein, as well as by a decrease in the expression of the tight junction (TJ) proteins, such as ZO-1 and CLDN-5 (Figure 3c). On the next day after the 1268 nm irradiation, when the BBB was closed, the BBB protein’s status changed. Indeed, the expression of Pgp, ZO-1 and CLDN-5 was restored to the control level (Figure 3c). The expression of MCT1, which mediates the transport of lactate across the BBB [62,63], decreased with the accumulation of lactate in the brain tissues (5.09 ± 0.01 nmol/mg vs. 7.33 ± 0.03 nmol/mg, *p* < 0.001, the Wilcoxon, Mann-Whitney U tests). In contrast, the expression of JAM and CD31 increased (Figure 3c).

One hour after the laser 1268 nm irradiation, TEER, reflecting the integrity of TJ proteins, decreased compared to the control group (Figure 3d). On the next day after the laser 1268 nm irradiation, TEER recovered and was similar to the control group.

Thus, the 1268 nm laser-induced BBBO is associated with a decrease in TEER, activation of the ABC-transport system via Pgp, as well as a temporal loss of the TJ (ZO-1 and CLND-5) presence in the endothelial gaps. The BBB recovery is accompanied by activation of neuronal metabolic activity (accumulation of lactate with a decrease in the expression of MCT1 by the mechanism of negative feedback), recovery of TEER, and stabilization of the BBB permeability with the elevation of expression of JAM and CD31 (Figure 3e).

### 3.5. The 1268 nm Laser-Induced BBBO as an Effective Technology for Brain drug Delivery

The GBM is characterized by high leakage of the blood vessels inside of the tumor mass, suggesting the high BBB permeability [19,64,65]. However, the blood vessels around GBM have IBBB, i.e., with the normal BBB permeability [18,20,66,67]. Note that the blood vessels surrounding the tumor are used by the GBM cells for migration and growth progression [66,67]. Therefore, the development of effective methods bypassing the IBBB at the border of GBM is the first step in the GBM therapy. Liposomes are promising nanocarriers to deliver anti-cancer drugs to GBM [37,68,69,70,71]. Therefore, in the final step, we studied the 1268 nm laser-induced BBBO for the delivery of liposomes into GBM. The interstitial fluid (ISF) flow is a crucial route of drug delivery into the brain [72]. However, brain tumors are associated with disorders of the ISF drainage [73] that sig-nificantly decrease the effectiveness of brain drug delivery and make it impossible to sustain adequate drug concentrations within the brain. Therefore, additionally, we studied whether the 268 nm laser can increase brain drainage in mice with GBM. Since anesthesia can affect the BBB permeability [74], we developed a portable technology for photo-influence on freely moving and non-anesthetized mice (Figure 1 and Figure 4d).

Figure 4a–c clearly demonstrate that the intensity of the fluorescent signal from liposomes was 5.4-fold higher in mice treated with the 1268 nm laser than in mice with-out the 1268 nm laser irradiation (53.5 ± 2.3 a.u. vs. 9.8 ± 1.1 a.u., *p* < 0.001, *n* = 7 in each group, the Wilcoxon, Mann-Whitney U tests). Furthermore, the laser 1268 nm significantly increased drainage of the brain (Figure 4e–j). Indeed, the distribution of FITCD in dorsal and in especially in ventral parts of the brain was significantly higher after the 1268 nm laser irradiation than in intact mice (dorsal part of the brain: 24.9 ± 2.3 a.u. vs. 12.7 ± 3.5 a.u., *p* < 0.01 and ventral part of the brain: 175.5 ± 4.8 a.u. vs. 6.8 ± 2.2 a.u., *p* < 0.001, *n* = 7 in each group, the Wilcoxon, Mann-Whitney U tests) (Figure 4j).

Thus, these results suggest that the laser 1268 nm-induced BBBO in the local part of the cortex is accompanied by a systemic activation of the brain drainage system that can contribute to a significant distribution of FITCD in the brain tissues and effective delivery of liposomes into the GBM tissues.

## 4. Discussion

In this pilot study, we discover an unknown mechanism of LLLT effects on the CNS. Transcranial infrared photostimulation is a non-pharmacological and non-invasive therapy for numerous brain diseases [75,76,77]. However, the laser 1268 nm-induced BBBO is shown for the first time in our in vivo (MRI and 2PLSM) and ex vivo (confocal imaging of the EBAC leakage using markers of pericytes and astrocytes) studies. This new generation laser induces direct oxygen photoexcitation, leading to a generation of ROS and ^1^O_2_ without the need for PS and with good penetration into the brain (up to 6 mm via the intact skull) [46,47,48,49]. In in vivo experiments using two lasers, 1268 nm (^1^O_2_/ROS generation) and 1122 nm (no ^1^O_2_/ROS generation), we clearly show that only the laser 1268 nm causes reversible BBBO but not the laser 1122 nm. In in vitro experiments using the BBB model and DHE, we reveal that the laser 1268 nm stimulates the ^1^O_2_ generation in the blood endothelial cells and astrocytes. Thus, these results suggest that the 1268 nm laser-induced ^1^O_2_ generation in the BBB elements can be a trigger mechanisms of the 1268 nm laser-mediated BBBO. This is a crucial addition to our previous studies, where we also showed that a single laser 1268 nm pulse triggers the dose-dependent ^1^O_2_ generation in both normal keratinocytes and tumor cells [46].

By studying the mechanisms, we find that the 1268 nm laser-induced BBBO is associated with a decrease in TEER, the activation of the ABC-transport system via Pgp, as well as a temporal loss of the TJ (ZO-1 and CLND-5) presence in the endothelial gaps. The BBB recovery is accompanied by the activation of neuronal metabolic activity (accumulation of lactate with a decrease in the expression of MCT1 by the mechanism of negative feedback), recovery of TEER, and stabilization of the BBB permeability with the elevation of expression of JAM and CD31. The increase in the level of lactate in the in vitro BBB model after the 1268 nm laser irradiation suggests that BBBO leads to an increase in the energy demand of astocytes that is covered by glucose from the blood by the glucose transporters in capillaries and brain cells [78].

These results are consistent with our previous data demonstrating the mechanisms of PDT (the laser 635 nm + 5-ALA)-induced BBBO [36,37,38,39,40]. The PDT-mediated BBBO is associated with a temporal decrease in the expression of TJ proteins, such as CLDN-5 and VE-cadherin as main components of the BBB integrity [37]. There is the hypothesis that PDT-BBBO causes temporal loss of the TJ proteins on the surface of the cerebral endothelium due to their internalization that can be mediated by signal molecules, such as the arrestin beta-1 and beta-2 [37,79,80]. The PDT can also affect directly the cerebral endothelium leading to increase in the gaps between the endothelial cells via the changes of cytoskeleton and vascular tone loss due to microtubule depolarization [81,82]. The oxidative stress is another possible mechanism underlying PDT-OBBB, as evidenced in the increase in the level of the malondialdehyde level in the mouse brain [37,83].

An interesting result is the fact that the laser 1268 nm induces BBBO in the local part of the cortex. However, the local BBBO is accompanied by a systemic activation of the brain drainage system that is associated with significant distribution of FITCD in the brain tissues and effective delivery of liposomes into the GBM tissues. We assume that the laser 1268 nm-induced BBBO stimulates drainage of ISF, which is an important route of brain drug delivery [72]. The BBBO-mediated increase in the ISF movement can be a reason of greater accumulation of liposomes in the GBM tissues in mice treated with the 1268 nm laser compared with intact animals. Notice that brain tumors are associated with disorders of the ISF drainage [73] that significantly decrease the effectiveness of brain drug delivery and make it impossible to sustain adequate drug concentrations within the brain. Our data demonstrating the mechanisms underlying BBBO and activation of the brain drainage system are presented in Refs. [39,41,84,85].

The study of mechanisms of photo-modulation of the brain drainage system is still in its infancy. In our previous studies, we demonstrated that BBBO is accompanied by activation of removal of unnecessary substances from the brain via the meningeal lymphatic vessels (MLVs) [39,41,84,85]. We observed a similar activation of the MLVs after sound-induced BBBO [84,85]. The BBBO-mediated activation of the MLVs might be explained by the expression of similar proteins in the lymphatic and blood vessel endothelium, including CLND-5, ZO-1, JAM, VE-cadherin, CD31 [86,87]. In various series of our experimental studies, we show that PDT-BBBO causes temporal changes in the TJs expression in both the blood and lymphatic vessel endothelium [37,88,89].

PDT-induced activation of the MLVs is associated with a PTD-related increase in the activity of endothelial nitric oxide (NO) synthase [90,91]. In series of experiments using the laser 1268 nm, it has been discovered an important role of NO in photo-induced dilation of the MLVs leading to an increase in lymphatic evacuation of red blood cells from the mouse brain with intraventricular hemorrhages [90]. Other mechanism of ^1^O_2_-mediated control of the vascular tone is the ability of ^1^O_2_ to regulate the blood pressure through formation of an amino acid-derived hydroperoxide [92].

The laser 1268 nm-BBBO can be a new strategy for effective brain drug delivery, including embedded in nanocarriers, in surrounding GBM places with IBBB [93]. It well known that GBM is characterized by the BBB disruption [64,65,66]. However, the blood vessels around GBM have the normal BBB permeability [66,67]. These blood vessels with IBBB, GBM uses for migration and growth progression [66,67]. Therefore, the laser 1268 nm-BBBO in the normal blood vessels can stimulate the development of effective methods bypassing IBBB at the border of GBM for the effective GBM therapy.

We also demonstrate a unique portable technology of automated laser effects on the BBB permeability in freely moving and non-anesthetized mice that is an important tool for the future studies of the mechanisms of BBBO by other methods. Note that anesthesia, which is widely used in the experiments, significantly affects the BBB permeability [74], which makes it difficult to correctly interpret scientific results.

## 5. Conclusions

In this study, we discover that the infrared laser (1268 nm) induces BBBO in the local part of the cortex that is accompanied by activation of the brain drainage system. These data indicate that the laser 1268 nm-induced BBBO stimulates the ISF drainage, which is an important route of brain drug delivery [72]. The BBBO-mediated increase in the ISF movement can lead to a stronger accumulation of liposomes in the GBM tissues that we observe in mice treated with the 1268 nm laser compared with intact animals. Furthermore, the laser 1268 nm-BBBO can be a new strategy for brain drug delivery in surrounding GBM places with IBBB for the effective GBM therapy. By studying the mechanisms, we uncover that the 1268 nm laser irradiation induces the ^1^O_2_ generation in the blood endothelial cells and astrocytes, which can be a trigger mechanism of BBBO. The laser-induced BBBO causes activation of the ABC-transport system with a temporal decrease in the expression of TJ proteins. The BBB recovery is accompanied by the activation of neuronal metabolic activity and stabilization of the BBB permeability. LLLT-BBBO can be used as a new opportunity of interstitial PS-free PDT for modulation of brain tumor immunity and improvement of immuno-therapy for GBM in small kids and infants in whom PDT with PSs, radio- and chemotherapy are strongly limited, as well as in adult patients with a high allergic reaction to PSs. The new results about LLLT-BBBO expose unknown the mechanisms of LLLT effects on the CNS.

## Figures and Tables

**Figure 1 pharmaceutics-15-00567-f001:**
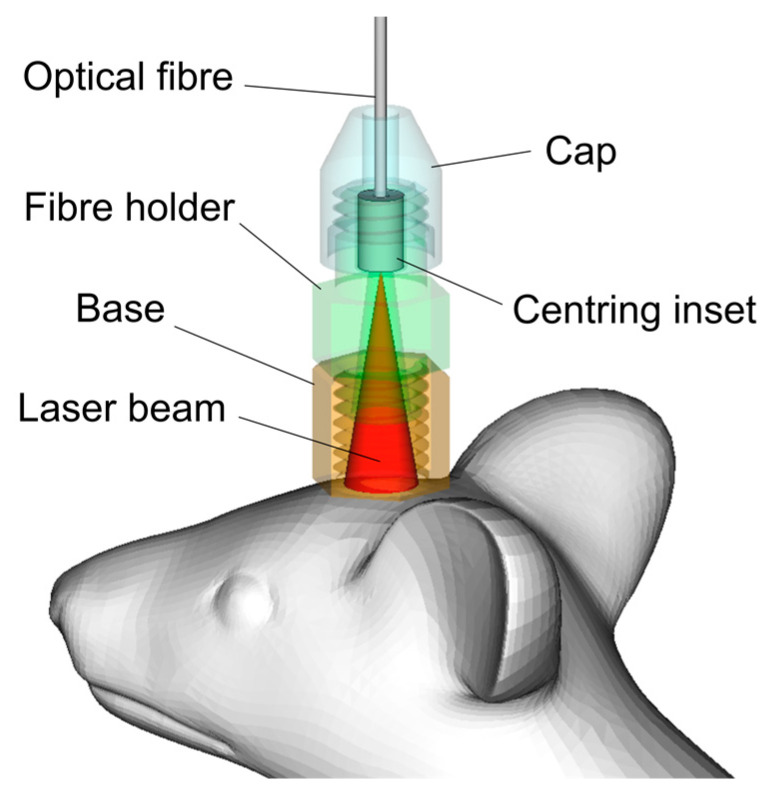
Schematic illustration of a device for the 1268 nm laser-induced BBBO in freely moving and non-anesthetized mice.

**Figure 2 pharmaceutics-15-00567-f002:**
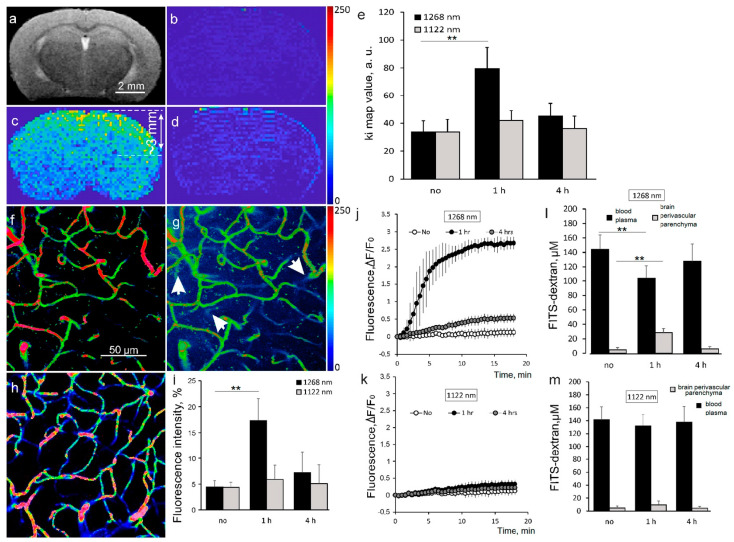
Singlet oxygen as a trigger for BBBO: (**a**)—The MRI data representing T2 (anatomical scan); (**b**)—The MRI data representing 15th rapid T1 MRI scan after the Gd-DTPA injection before the 1268 laser irradiation; (**c**)—The MRI data representing 15th rapid T1 MRI scan after the Gd-DTPA injection of the same brain 1 h after the 1268 laser irradiation; (**d**)—The MRI data representing 15th rapid T1 MRI scan after the Gd-DTPA injection 1 h after the 1122 laser irradiation; (**e**)—The Ki values (arbitrary units, a.u.) that is the rate of changes in the MRI signal intensity, reflecting the essential BBB leakage at 1 h after the 1268 nm laser irradiation with rapid the BBB recovery and no effect of the laser 1122 nm, *n* = 10, ** *p* < 0.01 vs. the basal value, the Mann–Whitney–Wilcoxon test; (**f**–**m**)—The in vivo 2PLSM of the BBB permeability for FITCD 70 kDa in mice subjected to laser irradiation with 1268 nm and 1122 nm. The analysis of microvascular permeability was performed before (no), 1 h and 4 h after the laser irradiation; (**f**)—The representative 2PLSM micrographs of mouse parietal cortex microvasculature 1 h after the laser irradiation with 1268 nm just after injection of FITCD; (**g**)—The same region at 20 min after FITCD injection, arrowheads point the FITCD leakage; (**h**)—The representative micrographs of microvasculature of the cortex irradiated with the laser 1122 nm at 20 min after the FITCD injection. No FITCD leakage was observed; (**i**)—The percentage fluorescence intensity in perivascular area of the brain tissues reflecting the maximum BBB leakage 1 h after the 1268 nm but not 1122 nm laser irradiation. Data are presented as mean ± SEM, *n* = 10, ** *p* < 0.01, the Mann–Whitney–Wilcoxon test; (**j**)—Time course of the FITCD extravasation showing its increased permeability 1 h after the 1268 nm irradiation. Data are presented as mean ± SEM, *n* = 10; (**k**)—Time course of the FITCD extravasation showing no significant changes after irradiation with the 1122 laser irradiation. Data are presented as mean ± SEM, *n* = 10; (**l**)—The estimation of the FITCD concentration in the brain perivascular parenchyma and in the blood plasma at 20 min after the FITCD injection and 1 h after the laser 1268 nm irradiation. Data are presented as mean ± SEM, *n* = 10, ** *p* < 0.01 vs. the basal value, the Mann–Whitney–Wilcoxon test; (**m**)—The estimation of the FITCD concentration in brain perivascular parenchyma and blood plasma at 20 min after the FITCD injection and 1 h after the laser 1122 laser irradiation. Data are presented as mean ± SEM, *n* = 10.

**Figure 3 pharmaceutics-15-00567-f003:**
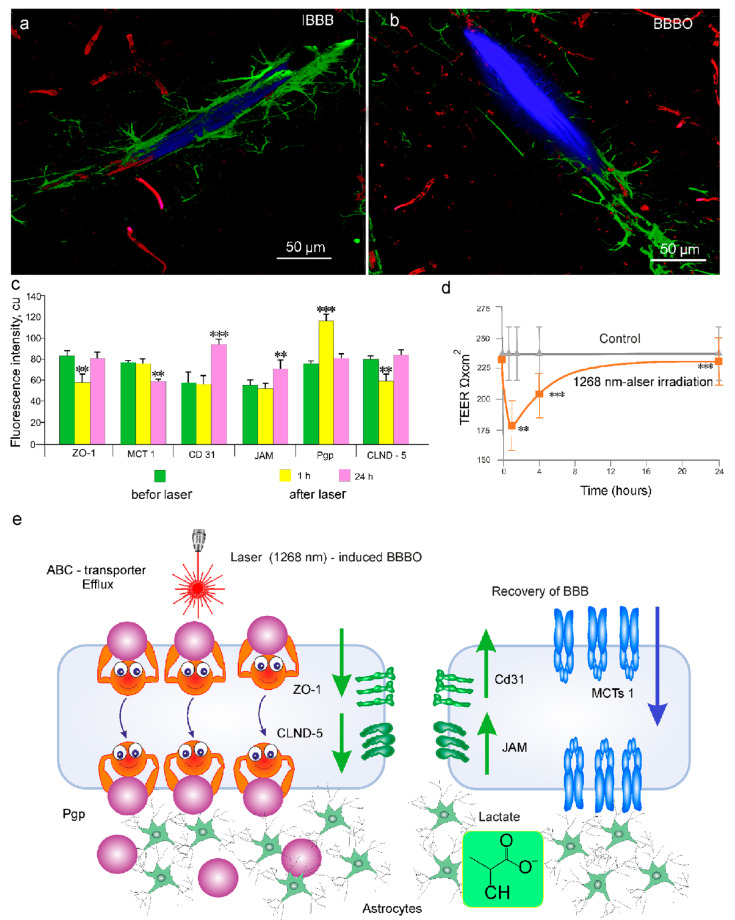
Mechanisms of underlying the 1268 nm-induced BBBO and recovery: (**a**,**b**)—Representative images of IBBB (**a**) and the 1268 nm laser-induced BBBO (**b**). The cerebral vessels are filled with the EBAC (blue), the pericytes (red), and the astrocytes (green) are marked with NG2 and GFAP, respectively. In the IBBB group, the EBAC is presented inside of the cerebral vessels; the EBAC leakage from the cerebral vessels into the brain parenchyma is observed in the BBBO group; (**c**)—The changes in the expression of the tested proteins (Pgp, CLDN-5, ZO-1, JAM, MCT1, CD31) in the cortex of the mouse brain without and 1 h or 24 h after the laser 1268 nm irradiation. **—*p* < 0.01; ***—*p* < 0.001 vs. the control group (no the laser irradiation), *n* = 10 in each group, the Wilcoxon, Mann-Whitney U tests; (**d**)—The laser 1268 nm-induced changes in the TEER in the BBB model, **—*p* < 0.01; ***—*p* < 0.001 vs. the control group (no laser irradiation), *n* = 10 in each group, the Wilcoxon, Mann-Whitney U tests; (**e**)—Schematic illustration of the mechanisms of laser-induced BBBO and recovery: The 1268 nm laser-induced BBBO is associated with a decrease in TEER, activation of the ABC-transport system via Pgp as well as a temporal loss of the TJ (ZO-1 and CLND-5) presence in the endothelial gaps. The BBB recovery is accompanied by activation of neuronal metabolic activity (accumulation of lactate with a decrease in the expression of MCT1 by the mechanism of negative feedback), recovery of TEER, and stabilization of the BBB permeability due to elevation of expression of JAM and CD31.

**Figure 4 pharmaceutics-15-00567-f004:**
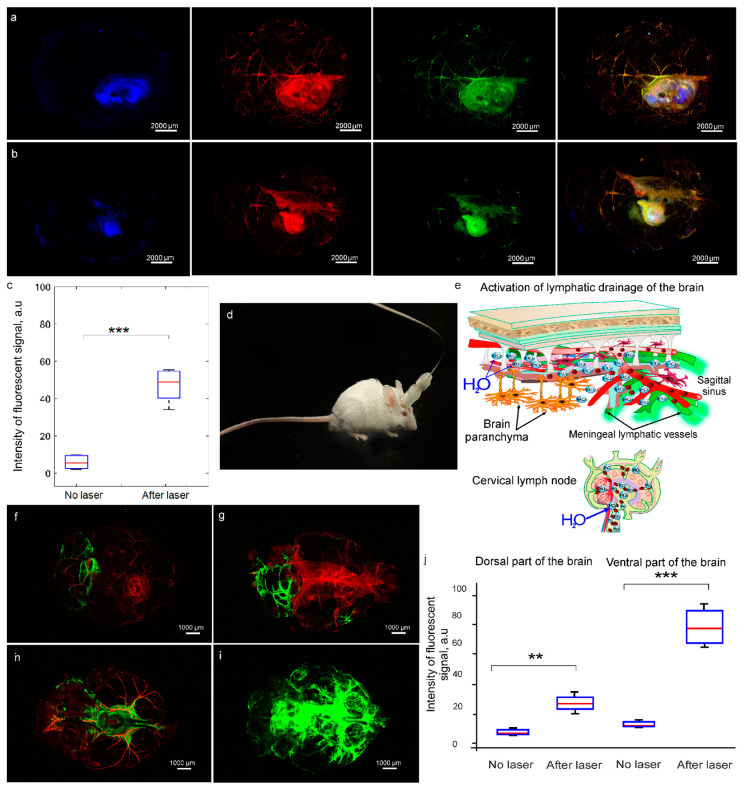
Delivery of liposomes into the GBM tissues and activation of the brain drainage system in mice without and after the laser 1268 nm-induced BBBO: (**a**,**b**)—Representative images of fresh samples of the whole brain demonstrating delivery of liposomes (green, Ex 561/Em 570–620) into the GBM tissues (blue, Ex 488/Em 500–550) in mice without (**a**) and after (**b**) the laser 1268 nm-induced BBBO. The cerebral vessels are filled with Evans Blue (red, Ex 640/Em 663-738); (**c**)—The quantitative analysis of the fluorescent signal from liposomes in the GBM tissues in mice without and after the laser 1268 nm-induced BBBO, ***—*p* < 0.001, *n* = 7, the Wilcoxon, Mann-Whitney U tests; (**d**)—Photo of a freely moving and non-anesthetized mouse with a device for the automated laser 1268 nm irradiation; (**e**)—Schematic illustration of the 1268 nm-laser-induced activation of the brain drainage system; (**f**–**i**)—Representative images of fresh samples of the whole brain demonstrating distribution of FITCD (green, Ex 488/Em 500-550) in dorsal (**f**,**g**) and ventral (**h**,**i**) parts of the brain without (**f**, **h**) and after (**g**, **i**) the laser 1268 nm irradiation. The cerebral vessels are filled with Evans Blue (red, Ex 640/Em 663–738); (**j**)—The quantitative analysis of the fluorescent signal from FITCD in the brain of mice without and after the laser 1268 nm-induced BBBO, ***—*p* < 0.001, **—*p* < 0.05, *n* = 7, the Wilcoxon, Mann-Whitney U tests.

## Data Availability

The data that support the findings of this study are available on request from the corresponding author.

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
