# Peer review of "Low-Level Laser Treatment Induces the Blood-Brain Barrier Opening and the Brain Drainage System Activation: Delivery of Liposomes into Mouse Glioblastoma"

_pharmaceutics, 2023, doi:10.3390/pharmaceutics15020567_

Round 1

Reviewer 1 Report

In this work, the authors tried to introduce a new method in opening BBB for drug delivery to brains, with the aim of better therapeutic results. Basically, this work is interesting, while the design is reasonable. In vitro and in vivo work have been done, with solid data. I believe this attempt should be encouraged in order to find more potential ways for brain drug delivery. However, there are still some parts that are not clear and should be clarified. I suggest a major revision after addressing the below points.

1, What does “a small application window (1-3 mm)” mean? This is confusing.

2, Ultrasound is so far the most successful technique in temporarily opening BBB. I don’t think it fair enough to raise several drawbacks of the method in order to stress the advantages proposed in this work. Compared with the ultrasound method, what is the most impressive achievement of laser?

3, To be honest, I hardly believe laser can replace even part of ultrasound-related work, since clinically some attempts have already been done with acceptable results. One of the most apparent shortcomings of laser, in my opinion, is still the penetration of laser due to its intrinsic natural (though NIR II light was used in this work). The skull drilling window could somehow solve the dilemma, however, might bring about more damage to patients. This work could be leading somewhere if the authors will dig more, for instance, the efficacy on lager animals (dogs or pigs), since laser penetration is greatly dependent on the volume of the targeted organs, as well as the current findings are basically based on mouse model and not so convincing, especially facing the clinical translation attempts.

4 Will some enduring damage to BBB will be caused after laser-induced PDT treatment?

5, It would be nicer if the authors could give some comments on the BBB penetration through employing the transporter (like GLUT-1) expressed on the epithelial cells of BBB. More related reference such as http://dx.doi.org/10.1016/j.apsb.2017.09.008 should be cited.

6, The longer of the wavelength of the lase, the higher of the penetration, meanwhile the lower of the energy and PDT capability. How to balance the design under this regard?

7, Have the authors compared with the current results with NIR-I light with shorter wavelength?

Author Response

Comments:

Ultrasound is so far the most successful technique in temporarily opening the BBB. I don’t think it fair enough to raise several drawbacks of the method in order to stress the advantages proposed in this work. Compared with the ultrasound method, what is the most impressive achievement of laser?

To be honest, I hardly believe laser can replace even part of ultrasound-related work, since clinically some attempts have already been done with acceptable results. One of the most apparent shortcomings of laser, in my opinion, is still the penetration of laser due to its intrinsic natural (though NIR II light was used in this work). The skull drilling window could somehow solve the dilemma, however, might bring about more damage to patients. This work could be leading somewhere if the authors will dig more, for instance, the efficacy on lager animals (dogs or pigs), since laser penetration is greatly dependent on the volume of the targeted organs, as well as the current findings are basically based on mouse model and not so convincing, especially facing the clinical translation attempts.

Response: The authors would like to express their sincere gratitude to the referee for constructive comments and suggestions to improve the quality of our article.

Thank you so much for these fair remarks. We completely agree with this. Therefore, we changed Title and re-wrote Introduction and Conclusion to make clearer the applied aspect of our data.

The results presented in our manuscript are logical continuation of series of our previous studies demonstrating PDT-mediated opening of the blood-brain barrier (BBBO) using the standard PDT protocol with laser 635 nm and 5-ALA [Biomed Opt Express 2017, 8, 5040–5048. doi: 10.1364/BOE.8.005040; Lasers Surg Med 2019, 51, 625-633. doi: 10.1002/lsm.23075; Biomed Opt Express 2018, 9, 4850–4862. doi: 10.1364/BOE.9.004850; J Biophotonics 2018, 11, e201700287. doi: 10.1002/jbio.201700287; J Biophotonics 2019, 12, e201800330. doi: 10.1002/jbio.201800330].

We also studied PDT-BBBO in mice using the different lasers and photosensitizers [Appl. Sci. 2020, 10, 33. https://doi.org/10.3390/app10010033].

PDT is an innovative and successful therapy of GBMs [J Neurosurg. 2020; 24:1-11. https://doi.org/10.3171/2019.11.JNS192443; https://clinicaltrials.gov/ct2/show/NCT01682746; Prog Neurol Surg 2018, 32, 1–13. doi: 10.1159/000469675; Front. in Surgery. 2020; 6:81-81. doi: 10.3389/fsurg.2019.00081].

The traditional explanation of the anti-cancer PDT effect is single oxygen-induced damage of the endothelial cells resulting in tumor cell death and microvasculature collapse. However, in our recent studies, the new vascular effects of PDT associated with BBBO and activation of the brain drainage system have been discovered [the references are presented above].

In our review, we discuss that PDT-BBBO is considered as the new niche in the development of innovative pharmacological strategies of modulation of brain tumor immunity and improvement of immuno-therapy for GBM [Pharmaceutics 2022, 14, 2612. https://doi.org/10.3390/pharmaceutics14122612].

We assume that the laser 1268 nm-induced BBBO stimulates the ISF drainage, which is an important route of brain drug delivery [Aging Dis. 2020, 11, 200-211. doi: 10.14336/AD.2020.0103].

In our recent study, we clearly demonstrate photo-stimulation of lymphatic delivery of liposomes into the GBM tissues [Pharmaceutics 2023, 15, 36. doi: 10.3390/pharmaceutics15010036].

The activation of the brain drainage system is a unique phenomenon, which we discover using other methods for BBBO [Computational and Structural Biotechnology Journal 21 (2023) 758–768 https://doi.org/10.1016/j.csbj.2022.12.019; Front. Oncol. 2022. 12:1010188. doi: 10.3389/fonc.2022.1010188; Proc R Soc B 2020, 28720202337. 20202337. https://doi.org/10.1098/rspb.2020.2337].

However, PDT has limitations to use in small children and infants as well as in adult patients with allergy to photosensitizers (PSs) [Journal of the German Society of Dermatology. 2021. https://doi.org/10.1111/ddg.14314; VIEW. 2023. 3. 20200121. https://doi.org/10.1002/VIW.20200121; J Dtsch Dermatol Ges. 2021 Jan;19(1):19-29. doi: 10.1111/ddg.14314; Pharmaceuticals (Basel). 2021 Jul 26;14(8):723. doi: 10.3390/ph14080723].

Therefore, the development of new PSs-free PDT can give new opportunities to overcome the limitations of PDT-therapy of GBM in such patients. In this aspect, recently developed compact and efficient quantum-dot lasers emitting in the near-infrared spectral range centered at around 1268 nm offer such a promising opportunity. This wavelength irradiation is capable to directly generate oxygen photoexcitation in complex media, leading to a generation initially singlet oxygen (1O2) and other reactive oxygen species without the need for PSs [Free Radic Biol Med 2021, 163: 306-313. https://doi.org/10.1016/j.freeradbiomed.2020.12.022].

We believe that the laser 1268 nm-induced BBBO can be used as a new opportunity of PS-free interstitial PDT (iPDT) for modulation of brain tumor immunity and improvement of immuno-therapy for GBM in small kids and infants in whom PDT with PSs, radio- and chemotherapy are strongly limited, as well as in adult patients with a high allergic reaction to PSs. The iPDT is a minimally invasive treatment of GBM that is used for therapy of recurrent GBM. iPDT is delivered with semiconductor lasers positioned inside the surgical cavity or directly into tumor mass [Cancers 2021, 13, 5754.  doi: 10.3390/cancers13225754].

Advantages of use laser treatment are selective cancer killing (Sci Rep 2013, 3(1), 3484. https://doi.org/10.1038/srep03484; Free Radic Biol Med 2021, 163: 306-313. https://doi.org/10.1016/j.freeradbiomed.2020.12.022), i.e. the laser 1268 nm induces suppression of the growth of tumor cells. Our results “Transcranial photosensitizer-free laser (1267 nm) treatment of glioblastoma in young rats” are under review in Neuro Oncology now.

The precision of laser treatment is significantly higher than focused ultrasound. Laser beam can be focused to sport ~1um, in case of ultrasound ~1mm which is 3 order of magnitude higher.

Comment: What does “a small application window (1-3 mm)” mean? This is confusing.

Response: We removed this phrase from the text.

Comment: Will some enduring damage to BBB will be caused after laser-induced PDT treatment?

Response: We presented in Figure 2SI the data of histological analysis of the brain tissues after the 1268 nm laser irradiation. Our results did not reveal damages of the brain tissues. We assume it is due to low-level laser application. However, PDT-BBBO with the high doses of laser 635 nm or 5-ALA is accompanied by injuries of the cerebral vessels leading to vasogenic edema [Biomed Opt Express 2017, 8, 5040–5048. doi: 10.1364/BOE.8.005040; Lasers Surg Med 2019, 51, 625-633. doi: 10.1002/lsm.23075; Biomed Opt Express 2018, 9, 4850–4862. doi: 10.1364/BOE.9.004850].

Comment: It would be nicer if the authors could give some comments on the BBB penetration through employing the transporter (like GLUT-1) expressed on the epithelial cells of BBB. More related reference such as http://dx.doi.org/10.1016/j.apsb.2017.09.008 should be cited.

Response: We added an explanation for increasing the level of lactate in the in vitro BBB model after the laser 1268 nm-induced BBBO as an increase in the energy demand of astocytes that is covered by glucose from the blood by glucose transporters in capillaries and brain cells (Lines 628-631). We cited the article (Lines 671-672).

Comment: The longer of the wavelength of the laser, the higher of the penetration, meanwhile the lower of the energy and PDT capability. How to balance the design under this regard?

Response: The selection of the laser wavelength was determined by the absorption spectrum of oxygen rather than the energy of quantum. A band of 40 nm centered at 1270 nm pumps aground state oxygen molecule into the first excited singlet state and is more advisable for direct singlet oxygen production in biological tissue. A comprehensive review on the direct singlet oxygen production can be found e.g. in [Blazquez-Castro, A. Redox biology, (2017) 13, 39-59. http://dx.doi.org/10.1016/j.redox.2017.05.011].

Comment: Have the authors compared with the current results with NIR-I light with shorter wavelength?

Numerous studies performed over previous decades have been demonstrated that the singlet oxygen production is possible at a series of bands down to UV, but the shorter wavelengths are less efficient because of less transition probability to the higher final vibrational levels of resulting singlet oxygen molecule [Blazquez-Castro, A. Redox biology, (2017) 13, 39-59. http://dx.doi.org/10.1016/j.redox.2017.05.011].

In our manuscript, we used two lasers 1122 nm, which does not produce singlet oxygen/ROS and 1268 nm, which generates singlet oxygen/ROS. Our in vivo MRI and 2PLSM results clearly demonstrate that the laser 1268 nm but not the laser 1122 nm induces reversible BBBO.

In our previous studies, we demonstrated the dose-related PDT-BBBO using the laser 635 nm + 5-ALA. However, here we show BBBO by singlet oxygen with PSs (5-ALA). In current study, we show BBBO by direct generation of singlet oxygen without PSs.

The authors would like to thank again the referee for the great help in improving our paper for its possible publication in Pharmaceutics.

Reviewer 2 Report

Generally, this manuscript was fairly written, but the rationale seems sound. Should the following issues be addressed, it is acceptable.

Figures need to be reorganized logically. Figs should appear in text in sequence. Fig 4d was not cited. Fig 3 appeared prior to fig 2 in the text. A careful inspection throughout the text is required.

More details, such as assay names, times,  doses, etc, should be given in the methods and individual captions. Whether fig 4a, 4b are fluorescent imaging of brain sections? What is the time point? How did you take the images?  With the same settings?

Line 99 Sown or seen?

Line 241 reorganize the sentence.

Line 255 degree symbol?

Figs, No means o? time units are not necessary for the horizon axels.  

Line 504 “no any” change to “not any”.  

Line 505, “1k” to “2k”.

Line 531, “3e” to “3d”.

Fog 3 caption, “BBBO(b)” to “BBBO(d)”. Fig 3e, before.

Author Response

Comments: Generally, this manuscript was fairly written, but the rationale seems sound. Should the following issues be addressed, it is acceptable. Figures need to be reorganized logically. Figs should appear in text in sequence. Fig 4d was not cited. Fig 3 appeared prior to fig 2 in the text. A careful inspection throughout the text is required.

Response: The authors would like to express their deep gratitude to the reviewer for the great help in improving our article and useful advices.

We improved organization of the figures in our manuscript and moved the results of analysis of the ROS production in the in vitro BBB model from the text of manuscript to the figure 1SI. Figure 4d is cited (Line 582).

Comments:

Line 99 Sown or seen?

Line 241 reorganize the sentence.

Line 255 degree symbol?

Figs, No means o? time units are not necessary for the horizon axels. 

Line 504 “no any” change to “not any”. 

Line 505, “1k” to “2k”.

Line 531, “3e” to “3d”.

Fog 3 caption, “BBBO(b)” to “BBBO(d)”. Fig 3e, before.

Response: All stylistic and grammatical errors have been corrected.

Comment: More details, such as assay names, times,  doses, etc, should be given in the methods and individual captions. Whether fig 4a, 4b are fluorescent imaging of brain sections? What is the time point? How did you take the images?  With the same settings?

Response: We added more details in the methods about optical monitoring of the brain drainage system and liposomes delivery into the GBM tissues (Lines 353-380).

“For the study of photo-activation of the brain drainage system, we investigated distribution of FITCD on dorsal and ventral parts of the brain after intraventricular injection of tracer in mice treated and not by the 1268 nm laser irradiation. Ten days before experiments, a polyethylene catheter (PE-10, 0.28 mm ID x 0.61 mm OD, Scientific Commodities Inc., Lake Havasu City, Arizona, United States) was implanted into the right lateral ventricle (AP - 1.0 mm; ML −1.4 mm; DV - 3.5 mm) for injection of FITCD according to the protocol reported by Devos et al. [61]. An amount of 5 μL of FITCD 70 kDa (Sigma-Aldrich, St Luis, USA) at a rate of 0.1 μL/min was injected into the right lateral ventricle in two groups of awake healthy mice, including animals treated with the 1268 nm laser irradiation and intact mice without the 1268 nm laser irradiation. The cerebral vessels were filled with the Evans Blue dye (2 mg∕100 g, 1% solution in physio-logical 0.9% saline, Sigma-Aldrich, St. Louis, MO, USA) that was injected intravenously via the tail vein. The FITCD and Evans Blue were injected 30 min before confocal and multiphoton microscopy.

To monitor accumulation of liposomes in the GBM tissue, liposomes in physiological 0.9% saline (0.2 ml/100 g, i.v.) were intravenously injected via the tail vein 30 min before confocal imaging [37].

The ex vivo optical visualization of the distribution of FITCD in the brains or accumulation of liposomes in the GBM tissues was performed with an A1R MP confocal microscopy system based on the Ni-E focusing nosepiece upright microscope (Nikon, Tokyo, Japan). Three laser excitation sources with 488 nm (for FITCD), 640 nm (for the Evans Blue dye) and 561 nm (for liposomes) wavelengths and a set of photomultipliers as detectors were used for multichannel confocal imaging with 2x and 10x objectives. Fresh samples of whole brains were immersed in saline solution in a Petri dish and covered with horizontally oriented cover glass. The Petri dish was placed on an electromechanical stage of the confocal imaging system. The images were obtained using NIS-Elements software (Nikon Instruments Inc., Tokyo, Japan) and analyzed using Fiji software (Open-source image processing software) and Vaa3D (Open Source visualization and analysis software)”. 

The authors thank once again the referee for the opportunity to improve the quality of our article for its possible publication in Pharmaceutics.

Round 2

Reviewer 1 Report

The reviewr is satisfied with this version.

Author Response

Comment: The review is satisfied with this version.

Response: The authors would like to thank the reviewer again for great support and important recommendations, which helped to significantly improve the quality of our manuscript for its possible publication in Pharmaceutics.

Authors

Reviewer 2 Report

Minor revision is still required. In fig 2, hours in x axel title might be deleted, and a big gap in the caption. 

Author Response

Comment: In fig 2, hours in x axel title might be deleted, and a big gap in the caption.

Response: The authors would like to thank the reviewer again for great help in improving the quality of our paper for its possible publication in Pharmaceutics.

We made a correction of figure 2 and its caption.

Authors
